# Ensemble Policy Optimization with Diversity-regularized Exploration

## Abstract

In machine learning tasks, ensemble methods have been widely adopted to boost the performance by aggregating multiple learning models. However, ensemble methods are much less explored in the task of reinforcement learning, where most of previous works only combine multiple value estimators or dynamics models and use a mixed policy to explore the environment. In this work, we propose a simple yet effective ensemble policy optimization method to improve the joint performance of the policy ensemble. This method utilizes a policy ensemble where heterogeneous policies explore the environment collectively and their diversity is maintained by the proposed diversity regularization mechanism. We evaluate the proposed method on continuous control tasks and find that by aggregating the learned policies into an ensemble policy in test time, the performance is greatly improved. DEPO has performance improvement and faster convergence over the base on-policy single-agent method it built upon. Code will be made publicly available.

## 1 Introduction

Ensemble methods, which train multiple learners to solve the same problem (Zhou, 2012), have been widely applied in machine learning to improve the model performance (Hansen & Salamon, 1990; Freund & Schapire, 1997; Dietterich, 2000; Singh et al., 2016; Huang et al., 2017). However, in the field of Reinforcement Learning (RL), ensemble methods are much less explored. So far previous works have studied the ensemble of value functions (Osband et al., 2016; Chen et al., 2017) or the ensemble of dynamics models (Rajeswaran et al., 2016; Chua et al., 2018; Buckman et al., 2018). Both works focus on fitting multiple estimators that are used to train the policy while the training data is still collected by a single policy that is derived from the ensemble of estimators. As a result, the exploration is bounded in a limited state-action subspace which is highly correlated to previous experience.

We consider adopting ensemble method in RL in its most straightforward form: improving the performance by averaging the action distributions of learned policies. We first examine an individual PPO policy ensemble, where we train 5 PPO policies independently. As shown in the Fig. 1, surprisingly, directly aggregating the independent policies turns out to be not effective. We find that those learned policies converge to different behavioral modes due to the initialization and their historical experience, thus simply averaging the outputs jeopardizes the performance.

To address above issue, we propose *Diversity-regularized Ensemble Policy Optimization (DEPO)*, a simple yet effective framework that augments the single-policy optimization with the power of ensemble method in policies. [1] Concretely, DEPO (1) trains multiple policies simultaneously by sharing data collected from each heterogeneous policy to maximize a novel *peer pressure objective*, (2) maintains the ensemble diversity via a non-parametric diversity regularizer, and (3) aggregates the policy ensemble to a mixture policy in test time, further boosting the performance over the well-trained individual policies. We benchmark our framework on the continuous control tasks. The experiments verify the effectiveness of the proposed ensemble policy optimization method in that DEPO outperforms the on-policy counterpart in final performance and

---

[1] Note that the term "ensemble policy optimization" is also used in (Rajeswaran et al., 2016), referred to the optimization of model ensemble, which is different to the policy ensemble described in this work.

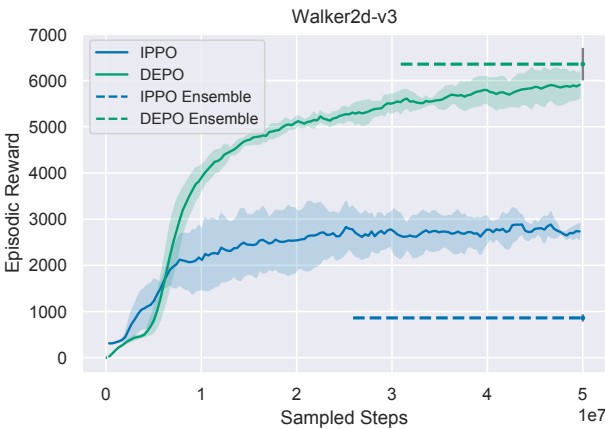

Figure 1: In this figure, `IPPO` refers to the experiment training 5 PPO policies independently. `DEPO` is our proposed learning framework where 5 PPO policies explore independently but are trained from shared data. The `Ensemble` results indicate the performance when the action is sampled from a uniform mixture of constituent policies in the ensemble $a \sim \sum_{i=1}^{K} \frac{1}{K} \pi_i(\cdot|s)$. DEPO can substantially improve the final performance. Noticeably the independently trained PPO policies yield inferior performance when using the mixed policy than the individual policy.

convergence speed. We also conduct detailed ablation studies to justify the techniques used in DEPO. Experiments on off-policy setting do not draw clear advantage of DEPO.

## 2 Related Work

**Ensemble method.** Apart from the ensemble methods commonly used in the classic machine learning tasks (Singh et al., 2016; Huang et al., 2017; Wen et al., 2020), an expanding body of works explores the ensemble method in Reinforcement Learning (RL). The ensemble methods in RL fall into roughly two categories: the ensemble of value functions and the ensemble of dynamics models in model-based RL.

The ensemble of value functions manages to reduce the variance of state value estimation (Hans & Udluft, 2010; Fujimoto et al., 2018), thus it can be used to encourage efficient exploration through selecting action by using upper-confidence bound (UCB) (Osband et al., 2016; Chen et al., 2017) or voting methods (Wiering & Van Hasselt, 2008; Faußer & Schwenker, 2011; Peng et al., 2016). In these methods, the behavior policy is retrieved from a mixture of Q functions. In offline RL, EDAC (An et al., 2021) uses an ensemble of Q networks to penalize OOD data with high uncertainties. However, the diversity encouragement in EDAC aims at learning better uncertainty estimation with less Q networks while, in DEPO, the diversity is encouraged for better exploration and final performance. In EDAC, the cosine similarity between the gradients of different Q networks are used as the diversity. In contrast, the proposed DEPO uses the difference action distribution of two policies on the same input data as the diversity reward.

On the other hand, the ensemble of dynamics models can mitigate the model approximation errors (Rajeswaran et al., 2016; Chua et al., 2018; Buckman et al., 2018) and accelerate policy learning by generating synthetic experiences from the ensemble (Kurutach et al., 2018). However, in most of the cases, there is only one training policy that exhibits in the system.

There are some works using multiple policies, but all in a centralized manner: there is only one "mixed" policy interacts with the environment. Both of Zheng et al. (2018) and Zhang & Yao (2019) maintain multiple critics in the system and use a voting method $a = \arg\max_{a_i} Q(s, a_i)$ to sample actions. The former work trains a set of independent critics separately and the latter maintains a centralized critic. Lee et al. (2020) proposes SUNRISE, which chooses action based on UCB (Chen et al., 2017) $a_t = \arg\max_a Q_{\text{mean}}(s_t, a) + \lambda Q_{\text{std}}(s_t, a)$ and trains each critic individually. Agarwal et al. (2020) and Misra et al. (2020) use historical policies to

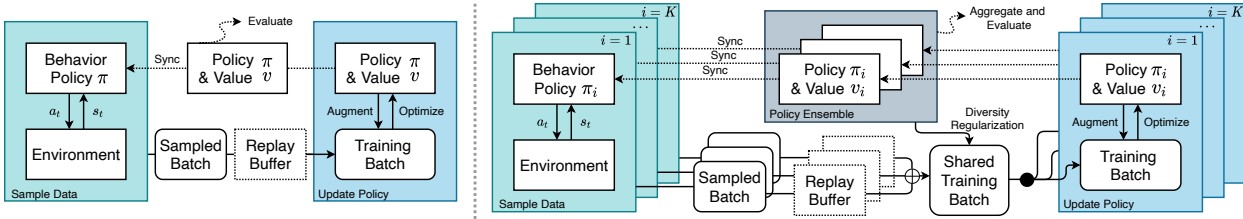

Figure 2: Compared to single policy optimization workflow, the proposed ensemble policy optimization framework incorporates multiple heterogeneous policies to execute in independent environments and share the data among the policy ensemble during training. The diversity regularization mechanism further preserves the ensemble diversity.

optimize current policy for global optimal policy via larger state-action space coverage. Instead, we use a policy ensemble where each constituent policy explores the environment independently at the same time.

**Exploration with shared experience.** A similar domain to the policy ensemble method is the distributed RL, where large quantities of parallel actors collect data simultaneously. Mnih et al. (2016) propose A3C and A2C that use multiple independent actors to explore the environments. Both methods maintain a global agent who receives the gradients from parallel actors and episodically broadcasts the latest parameters of the global policy to actors. IMPALA (Espeholt et al., 2018) is similar to A3C but is different in mixing samples instead of gradients from actors and updating the global policy based on the shared samples. Compared to A3C, A2C, and IMPALA, DEPO maintains a group of heterogeneous policies and eliminates the concept of a centralized policy during training.

Besides, our work is different from the works on Multi-agent RL where multiple agents interact with each others in the same environment (Lowe et al., 2017; Gupta et al., 2017; Rashid et al., 2018). DEPO focuses on improving the exploration with policy ensemble in single agent tasks. The different agents explore in independent environments and have no influence on each other during the sampling period.

**Learning with diversity.** In deep RL community, many works encourage diversity explicitly through adding extra loss to make an agent behave differently (Hong et al., 2018; Masood & Doshi-Velez, 2019); creating conjugate policies which improve the main policy by adding noise in parameters (Cohen et al., 2019); or adding diversity as explicit reward (Eysenbach et al., 2018). Zhang et al. (2019) propose a method called task novelty bisector (TNB) which boosts the diversity with gradient fusion. The diversity regularizer in DEPO is different in the following aspects: (1) It trains multiple policies simultaneously, instead of generating a population of policies sequentially; (2) DEPO uses a simple and computationally efficient form of diversity, which does not require extra auto-encoder (Burda et al., 2018), hand-craft behavioral representation (Mouret & Clune, 2015), or complex heuristics to balance the task and diversity objectives (Hong et al., 2018).

## 3 Method

### 3.1 Preliminary

Reinforcement learning methods usually follow the single-policy optimization setting, where an agent explores the environment to learn optimal behavior that can maximize the expected return:

$$J(\pi_\theta) = v_{\pi_\theta} = \mathop{\mathbb{E}}_{\pi_\theta} \sum_{t=0} \gamma^t r(s_t, a_t), \tag{1}$$

wherein $s_t$, $a_t$, $\gamma$ and $r$ are the state, action, discount factor and reward respectively. $\pi_\theta$ is the policy parameterized by $\theta$. $v_{\pi_\theta} = \mathbb{E}_{s_0} v_{\pi_\theta}(s_0)$ is the expected return, wherein $v_\pi(s_t) = \mathbb{E}_\pi \sum_{t'=t} \gamma^{(t'-t)} r(s_{t'}, a_{t'})$ is the state value.

The policy gradient methods are typical on-policy RL algorithms. These methods derive the policy gradient (Sutton & Barto, 2018) $\nabla_\theta J(\pi_\theta)$ following:

$$A_{\pi_\theta}(s_t, a_t) = \quad r(s_t, a_t) + \gamma v_{\pi_\theta}(s_{t+1}) - v_{\pi_\theta}(s_t), \tag{2}$$

$$\nabla_\theta J(\pi_\theta) = \quad \mathbb{E}_{(s,a) \sim P_{\pi_\theta}}[\nabla_\theta \log \pi_\theta(a|s)\hat{A}_{\pi_\theta}(s,a)], \tag{3}$$

where $(s, a)$ is collected from the exploration of policy $\pi_\theta$ and $P_{\pi_\theta}$ denotes the data distribution. The advantage $A_{\pi_\theta}(s_t, a_t)$ describes the relative improvement of action $a_t$ over the *baseline*, namely the state value $v_{\pi_\theta}(s_t)$, at step $t$. Eq. 3 can be obtained by differentiating the surrogate objective:

$$J(\pi_\theta) = \mathbb{E}_{(s,a) \sim P_{\pi_\theta}} [\log \pi(a|s)\hat{A}_\pi(s,a)], \tag{4}$$

wherein $P_{\pi_\theta}$ is the state-action visitation distribution deduced by policy $\pi_\theta$. Applying the policy gradient to policy parameters with stochastic gradient ascent can improve the expected return.

In popular RL method PPO (Schulman et al., 2017), a clipped surrogate objective is used to modulate the distributional shift between behavior policy $\pi_{\text{old}}$ and target policy $\pi_{\text{new}}$, so that it can update the target policy multiple times with same data collected by $\pi_{\text{old}}$:

$$J_{\text{PPO}}(\pi_{\text{new}}) = \mathbb{E}_{(s,a) \sim P_{\pi_{\text{old}}}} [\min(\rho\hat{A}_{\pi_{\text{old}}}, \text{clip}(\rho, 1 - \epsilon, 1 + \epsilon)\hat{A}_{\pi_{\text{old}}})], \tag{5}$$

$\rho$ denotes the probability ratio $\rho = \pi_{\text{new}}(s, a)/\pi_{\text{old}(s,a)}$, and $\epsilon > 0$ is the clipping parameter. The ratio clipping can reduce the variance of policy gradient estimation and constrains $\pi_{\text{new}}$ remaining close to $\pi_{\text{old}}$.

As illustrated in left panel of Fig. 2, we can summarize the learning pipeline of single policy optimization methods as following four steps: (1) Sampling: a behavior policy interacts with environments and collects a sampled batch, (2) Augmentation: the samples are augmented with value targets or advantages and formed into a batch, (3) Updating: the policy is updated based on the augmented training batch. (4) Synchronization: update the behavior policy according to the latest target policy.

### 3.2 Ensemble Policy Optimization Framework

Extending the single-policy optimization, we consider a policy optimization framework which supports multiple heterogeneous policies running in parallel exploring the same task. The ensemble policy optimization (EPO) consists three components: (1) a *policy ensemble* that contains K policies, each policy interacts with the environment independently to roll-out a batch of samples; (2) a *training algorithm* $\mathcal{T}$ that updates the ensemble and should be capable to integrate data collected from each policy; and (3) an *aggregation function* $\mathcal{G}$ that proposes a mixed policy at test time $a_t \sim \mathcal{G}(\{\pi_i\}_{i=1}^K)$ that further boosts the performance.

We instantiate the EPO framework with *Diversity-regularized Ensemble Policy Optimization (DEPO)*. DEPO maintains a policy ensemble that contains $K$ policy networks without weight-sharing among them. As illustrated in the right panel of Fig. 2, during sampling period, DEPO requests each policy to roll-out a *sampled batch* containing $n = N/K$ transitions from the environment, wherein $N$ denotes the size of sampled batch in single policy optimization. DEPO then forms a *shared training batch* with totally $N$ samples by concatenating transitions $\{(s_t, a_t, r_t, s_{t+1}) \sim P_{\pi_i}\}_{i=1}^K$ from all $K$ policies. The shared batch is dispatched to each policy's optimizer, augmented by the target policy and value estimator, and then is used to optimize neural networks of the target policy.

In test time, DEPO aggregates the policy ensemble with a uniform mixture of all policies:

$$a \sim \mathcal{G}(\cdot|s, \pi_1, ..., \pi_K) = \frac{1}{K}\sum_{i=1}^K a_i, a_i \sim \pi_i(\cdot|s). \tag{6}$$

Note that to maximize the diversity of the sampled data, we do not aggregate policies during training. There are multiple behavior policies exploring each environment independently. This is the key difference to previous works. In the next section, we will discuss how DEPO trains each policy in the ensemble.

### 3.3 Peer Pressure Objective

We define the *peer pressure objective* for arbitrary policy $\pi_k$ (a shorthand of $\pi_{\theta_k}$) by slightly changing the Eq. 1 as:

$$J_{\text{PP}}(\pi_k) = v_{\pi_k} - \frac{1}{K}\sum_{i=1}^{K} v_{\pi_i}. \tag{7}$$

This objective incentivizes the agent to maximize its expected return while outperforming the averagely performing policies in the ensemble. In following, we will derive a practical optimization objective such that we can effectively utilize the data sampled by heterogeneous policies. We first revisit the following lemma:

**Lemma 1 (The performance difference lemma)** *For arbitrary policies $\pi_k$ and $\pi_i$, their expected performance difference is:*

$$v_{\pi_k} - v_{\pi_i} = \frac{1}{1-\gamma} \mathop{\mathbb{E}}_{s\sim P_{\pi_k}} \mathop{\mathbb{E}}_{a\sim\pi_k} [A_{\pi_i}(s,a)]. \tag{8}$$

This lemma is proved by Kakade & Langford (2002). We use this lemma to decompose the peer pressure objective. After applying this lemma, Eq. 7 can be written as:

$$
\begin{aligned}
J_{\text{PP}}(\pi_k) &= \frac{1}{K(1-\gamma)}\sum_{i=1}^{K} \mathop{\mathbb{E}}_{s\sim P_{\pi_k}} \mathop{\mathbb{E}}_{a\sim\pi_k} [A_{\pi_i}(s,a)] \\
&= \frac{1}{K(1-\gamma)}\sum_{i=1}^{K} \mathop{\mathbb{E}}_{s\sim P_{\pi_k}} \mathop{\mathbb{E}}_{a\sim\pi_i} [\rho(\pi_k,\pi_i)A_{\pi_i}(s,a)],
\end{aligned} \tag{9}
$$

wherein $\rho(\pi_k,\pi_i) = \pi_k(a|s)/\pi_i(a|s)$ is the importance sampling coefficient. The peer pressure objective $J_{\text{PP}}(\pi_k)$ queries the action distributions as well as the advantages of policy $\pi_i$ under the state distribution deduced by policy $\pi_k$. However, suppose we replace the state distribution $s \sim P_{\pi_k}$ by $P_{\pi_i}$, then we can directly utilize the transitions sampled by the policy $\pi_i$ without replaying $\pi_i$ on the dataset collected by $\pi_k$. We will justify such approximation in the following Theorem 1. We first write the *DEPO objective* for training policy $\pi_k$ as:

$$J_{\text{DEPO}}(\pi_k) = \frac{1}{K}\sum_{i=1}^{K} \mathop{\mathbb{E}}_{(s,a)\sim P_{\pi_i}} \rho(\pi_k,\pi_i)\hat{A}_{\pi_i}(s,a). \tag{10}$$

*The DEPO objective is a practical form of $J_{PP}$ that can be easily computed using the shared training batch.* In the following theorem, we state that DEPO objective approximates the peer pressure objective.

**Theorem 1** *The peer pressure objective in Eq. 7 is bounded by DEPO objective:*

$$J_{DEPO}(\pi_k) - D \leq J_{PP}(\pi_k) \leq J_{DEPO}(\pi_k) + D, \tag{11}$$

*wherein $D$ describes the divergence between the target policy $\pi_k$ and others as*

$$D = \frac{4\gamma}{K(1-\gamma)^2}\sum_{i=1}^{K} \epsilon_i \max_s D_{KL}(\pi_i(\cdot|s)||\pi_k(\cdot|s)), \tag{12}$$

*where $\epsilon_i = \max_{(s,a)} |A_{\pi_i}(s,a)|$.*

To prove this theorem, we first introduce a lemma according to the Theorem 1 in (Schulman et al., 2015).

**Lemma 2** *Define a function:*

$$L_{\pi_i}(\pi_k) = v_{\pi_i} + \mathop{\mathbb{E}}_{s\sim P_{\pi_i}} \mathop{\mathbb{E}}_{a\sim\pi_k(\cdot|s)} \hat{A}_{\pi_i}(s,a). \tag{13}$$

*Then the following inequality holds:*

$$|v_{\pi_k} - L_{\pi_i}(\pi_k)| \leq \frac{4\gamma\epsilon_i}{(1-\gamma)^2}D_{KL}^{\max}(\pi_i,\pi_k), \tag{14}$$

*where $\epsilon_i = \max_{(s,a)} |A_{\pi_i}(s,a)|$, and $D_{KL}^{\max}(\pi_i,\pi_k) = \max_s D_{KL}(\pi_i(\cdot|s)||\pi_k(\cdot|s))$.*

Note that the authors of Schulman et al. (2015) only give the lower bound of $v_{\pi_k} - L_{\pi_i}(\pi_k)$ but they actually proved the upper bound at the same time.

For simplicity, define $D_i = \frac{4\gamma\epsilon_i}{(1-\gamma)^2} D_{\mathrm{KL}}^{\max}(\pi_i, \pi_k)$. Therefore $D = \frac{1}{K}\sum_{i=1}^{K} D_i$. Following Eq. 14, we have:

$$\mathop{\mathbb{E}}_{\substack{s\sim P_{\pi_i}\\a\sim\pi_k(\cdot|s)}} \hat{A}_{\pi_i}(s,a) - D_i \le v_{\pi_k} - v_{\pi_i} \le \mathop{\mathbb{E}}_{\substack{s\sim P_{\pi_i}\\a\sim\pi_k(\cdot|s)}} \hat{A}_{\pi_i}(s,a) + D_i \tag{15}$$

Now we build the connection between $J_{\mathrm{DEPO}}$ and $J_{\mathrm{PP}}$. Since $\mathbb{E}_{a\sim\pi_k} \hat{A}_{\pi_i}(s,a) = \mathbb{E}_{a\sim\pi_i} \rho(\pi_k,\pi_i)\hat{A}_{\pi_i}(s,a)$, we have:

$$J_{\mathrm{DEPO}}(\pi_k) = \frac{1}{K}\sum_{i=1}^{K} \mathop{\mathbb{E}}_{\substack{s\sim P_{\pi_i}\\a\sim\pi_k(\cdot|s)}} \hat{A}_{\pi_i}(s,a) \tag{16}$$

Compute the average of Eq. 15 over all $i$s, we have:

$$\begin{aligned}
J_{\mathrm{DEPO}}(\pi_k) - D &= \frac{1}{K}\sum_{i=1}^{K}[\mathop{\mathbb{E}}_{\substack{s\sim P_{\pi_i}\\a\sim\pi_k(\cdot|s)}} \hat{A}_{\pi_i}(s,a) - D_i]\\
&\le \frac{1}{K}\sum_{i=1}^{K}[v_{\pi_k} - v_{\pi_i}] = J_{\mathrm{PP}}(\pi_k)\\
&\le \frac{1}{K}\sum_{i=1}^{K}[\mathop{\mathbb{E}}_{\substack{s\sim P_{\pi_i}\\a\sim\pi_k(\cdot|s)}} \hat{A}_{\pi_i}(s,a) + D_i] = J_{\mathrm{DEPO}}(\pi_k) + D
\end{aligned} \tag{17}$$

The main theorem is proved. Note that $D$ measures the divergence between $\pi_k$ and others policies. Since we update policy $\pi_k$ with the shared data batch, D will naturally reduce after each training iteration because all policy are updated to maximize the log action probability of the same set of high-return actions. Therefore $J_{\mathrm{DEPO}}$ approximates $J_{\mathrm{PP}}$.

### 3.4 Learning Objectives

To bound the policy update and reduce variance, we further apply the clipped surrogate objective as in Eq. 5 to the DEPO objective in Eq. 10. However, we find the clipped objective leads to numerical instability and will catastrophically collapse the learning since the objective is not bounded when the advantage is negative, as the PPO Loss shown in Fig. 3. This phenomenon is also noticed in (Ye et al., 2019). To tackle this issue, we use a Two-side Clip (TSC) surrogate objective to mitigate the variance of advantage as shown by the TSC Loss in Fig. 3. Concretely, the TSC loss equipped by DEPO is computed as follows:

$$J_{\mathrm{TSC}}(\pi_k) = \frac{1}{K}\sum_{i=1}^{K} \mathop{\mathbb{E}}_{(s,a)\sim P_{\pi_i}} [\mathrm{clip}(\rho(\pi_k,\pi_i), 0, 1+\epsilon)\hat{A}_{\pi_i}(s,a)]. \tag{18}$$

The ablation studies in Sec. 4.4 shows that data sharing across ensemble can already boost the performance. In next section, we will discuss an important problem brought by the data sharing.

### 3.5 Diversity Regularization

Due to the data sharing among policies, it is inevitable that all the constituent policies gradually become identical during the course of learning. To further improve the effectiveness of DEPO, we propose the Diversity Regularization (DR) mechanism, which regularizes the exploration and preserves the diversity of policies.

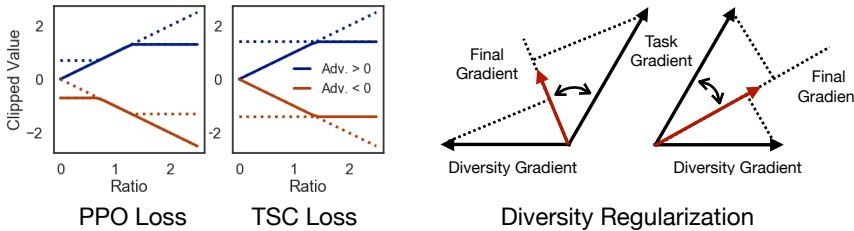

Figure 3: Illustration of the PPO loss, Two-side Clip loss, and the gradient fusion method in diversity regularization.

Considering the continuous control tasks we focus on, we use the Mean Square Error (MSE) between the means of action distributions produced by two agents as the diversity reward Hong et al. (2018). MSE is bounded since each action dimension is limited to $[-1, 1]$, avoiding unbounded value in other metrics like KL divergence. Denoting $\mu_k(s)$ as the mean of a stochastic action distribution $\pi_k(\cdot|s)$, MSE can be written in the closed form $||\mu_k - \mu_i||_2^2$. We therefore use the following diversity reward:

$$r_d^{(k)}(s) = \frac{1}{K-1} \sum_{i=1,i\neq k}^{K} ||\mu_k(s) - \mu_i(s)||_2^2. \tag{19}$$

We do not treat the diversity reward as intrinsic reward, instead, we use the diversity reward to compute the diversity objective to update the policy following Eq. 5 and then form *diversity gradient* based on such objective. The diversity gradient is fused with the primal task gradient later. The ablation study on using the intrinsic reward method is presented in Sec. 4.4.

DR retrieves the final gradient from two streams of gradients using the Feasible Direction Method (FDM). As illustrated in Fig. 3, we flatten the gradients w.r.t. all parameters into a vector for both objectives and get the task gradient and diversity gradient $\mathbf{g}_t, \mathbf{g}_d \in \mathbb{R}^{|\theta|}$ respectively. Then we compute the angular bisector of two flattened gradients in the parameter space: $\mathbf{d} = \mathcal{Z}(\mathcal{Z}(\mathbf{g}_t) + \mathcal{Z}(\mathbf{g}_b))$, wherein $\mathcal{Z}(\mathbf{x}) = \mathbf{x}/||\mathbf{x}||_2$ normalizes input to a unit vector. $\mathbf{d}$ therefore is a unit vector representing direction of the final gradient. We project two gradient vectors into $\mathbf{d}$ and use the average as the magnitude of final gradient. The final gradient after fusion is computed as follows:

$$\mathbf{g}_{\text{final}} = \frac{\mathbf{g}_t \cdot \mathbf{d} + \min(\mathbf{g}_d \cdot \mathbf{d}, \mathbf{g}_t \cdot \mathbf{d})}{2} \cdot \mathbf{d}. \tag{20}$$

min operation ensures the diversity reward does not prevail against the task reward.

The final gradient direction $\mathbf{d}$ is the angular bisector ensuring there always exist non-negative components of both gradient after the projection $\mathbf{g}_t \cdot \mathbf{d}$ and $\mathbf{g}_d \cdot \mathbf{d}$. The bisector therefore can improve both objectives effectively (Zhang et al., 2019). An alternative is to simply add two gradient vectors (Hong et al., 2018). However, such approach introduces a trade-off factor that is hard to tune. Evaluation in Sec. 4.4 shows that the FDM method performs better than using auxiliary loss.

To reduce the variance when estimating diversity across ensemble and possible trajectories, we introduce a diversity value network (DVN) to estimate the diversity value and compute the diversity gradient. DVN updates to minimize the Bellman error between the predicted values and $r_d^{(k)}(s) + \gamma DVN(s_{t+1}, a_{t+1})$. We compute the diversity in online manner, so the diversity reward is non-stationary during the training. To relieve such issue, we invite a common trick called *Delayed Update Target* (Lillicrap et al., 2015). We maintain a set of target policies by Polyak averaging the parameters of the policy ensemble over the course of training:

$$\theta_{\text{target}}^{(k)} \leftarrow (1-\tau)\theta_{\text{target}}^{(k)} + \tau\theta_{\text{latest}}^{(k)}, \quad \forall k = 1, ..., K, \tag{21}$$

wherein $0 < \tau \leq 1$ is a hyper-parameter. We then compute the diversity of a given policy $\pi_k$ against such target policies, including the delayed update target of $k$-th policy itself.

# 4 Experiments

## 4.1 Setup

We implement DEPO using RLLib (Liang et al., 2018). Generally, we host 8 concurrent trials in an Nvidia GeForce RTX 2080 Ti GPU. Each trial consumes 3 CPUs with 10 parallel rollout workers. To ensure efficiency, we hosts the sampling and training pipelines of each policy in separate workers running in parallel. DEPO only synchronizes when sharing the data and computing diversity. *The wall-time overhead therefore is trivial compared to single-policy methods.*

To ensure fair comparison, we ensure (1) the total number of interactions with environments and (2) the total number of sampled transitions in each training iteration for the whole DEPO system to be identical to the single-policy methods. Concretely, we set the total number of sampled steps to $5 \times 10^7$ for on-policy baselines and DEPO. In each training iteration, the whole system of DEPO collects $10,000$ transitions. For DEPO ensemble with 5 policies, this means each policy has the quota of $2,000$ steps to interact with its environment. All 5 policies will form a shared training batch with totally $10,000$ transitions, equal to the size of the training batch in on-policy single-policy baselines.

For all experiments, we use fully-connected neural networks with two layers, 256 hidden units per layer, for both policy networks and value networks. We use ReLU as the activation function. Other hyper-parameters in both on-policy and off-policy setting are listed in Appendix.

The implementation of OAC follows the code provided by the original paper (Ciosek et al., 2019). Note that the total training timesteps of this work is different from OAC paper. We use the official implementation of Actor-critic Ensemble (Zhang & Yao, 2019), SUNRISE (Lee et al., 2020), and TD3 (Fujimoto et al., 2018). The PPO, A2C, APPO and SAC implementations follow RLLib (Liang et al., 2018).

We evaluate methods in five continuous control locomotion tasks: HalfCheetah-v3, Ant-v3, Walker2d-v3, Hopper-v3, and Humanoid-v3 in MuJoCo simulator (Todorov et al., 2012). All experiments are repeated 5 times with different random seeds and the standard deviation of the values are presented in tables as well as the shadow of curves. The ensemble size is $K = 5$ if not explicitly stated.

## 4.2 Main Results

To validate that the ensemble policy optimization can improve the performance over single policy schemes, we compare DEPO with following baselines:

- `On-policy`: We compare with `A2C` (Mnih et al., 2016), `PPO` (Schulman et al., 2017), and `APPO`, a variant to IMPALA (Espeholt et al., 2018). In our preliminary experiments, we find A3C and IMPALA are unstable and sometimes fail the training, so we instead use A2C, the synchronized version of A3C, and APPO, which replaces the V-trace loss in IMPALA with the surrogate loss in PPO but still using the asynchronized infrastructure proposed in RLLib.

- `Exploration`: One exploration-enhanced method is benchmarked: `TNB` (Zhang et al., 2019). TNB aims at seeking diverse policies and trains a population of polices sequentially so the time consumption is much larger than ours.

As shown in Fig. 4, in all five tasks, our method (`DEPO`) outperforms the single-policy baseline with a large margin. Using the aggregated mixture policy in test time yields even more powerful policies that further boosts the performance (`DEPO Avg.`) compared to individual policies. In Hopper-v3, PPO collapses after a long time of training, while DEPO maintains its performance, which shows that DEPO is stable during training. In Table 1, we can see that DEPO outperforms on-policy baselines in all environments. We also implement off-policy DEPO and compare with various off-policy baselines. Please refer to Appendix for details.

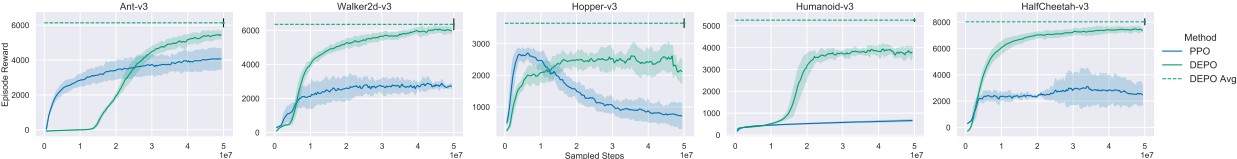

Figure 4: The learning curves of the PPO and on-policy DEPO in five environments. The performance of policy in DEPO ensemble (`DEPO`) already outperforms baselines in all tasks. Using the mixture policy of the ensemble in test time further boosts the performance (`DEPO avg.`).

Table 1: The episodic reward of baselines and proposed framework. `Elite` refers to the best policy in the ensemble and `Average` is the performance of the mixture policy of DEPO ensemble. DEPO outperforms all on-policy baselines and achieves competitive performance compared to powerful off-policy, exploration and ensemble method baselines.

| Category | Method | Ant-v3 | HalfCheetah-v3 | Hopper-v3 | Humanoid-v3 | Walker2d-v3 |
|---|---|---|---|---|---|---|
| Proposed Framework | DEPO Average | $6107.2_{\pm 258.9}$ | $8022.8_{\pm 293.1}$ | $3608.0_{\pm 135.8}$ | $5276.0_{\pm 61.0}$ | $6379.4_{\pm 241.1}$ |
| | DEPO Elite | $5487.6_{\pm 239.8}$ | $7529.5_{\pm 210.8}$ | $2922.8_{\pm 288.5}$ | $4114.0_{\pm 136.9}$ | $6179.1_{\pm 271.6}$ |
| On-policy Baseline | PPO | $4155.4_{\pm 596.9}$ | $3559.4_{\pm 1041.1}$ | $2860.1_{\pm 166.2}$ | $663.4_{\pm 55.8}$ | $3209.8_{\pm 280.1}$ |
| | APPO | $1460.9_{\pm 427.7}$ | $2814.6_{\pm 75.8}$ | $2038.1_{\pm 421.6}$ | $1351.9_{\pm 32.9}$ | $3159.9_{\pm 187.5}$ |
| | A2C | $1881.8_{\pm 185.4}$ | $2882.7_{\pm 1293.2}$ | $2132.9_{\pm 97.1}$ | $519.9_{\pm 90.5}$ | $1891.3_{\pm 664.6}$ |
| Exploration | TNB | $2211.0_{\pm 250.3}$ | $1623.6_{\pm 94.6}$ | $2916.8_{\pm 383.2}$ | $493.2_{\pm 40.8}$ | $2906.3_{\pm 152.3}$ |

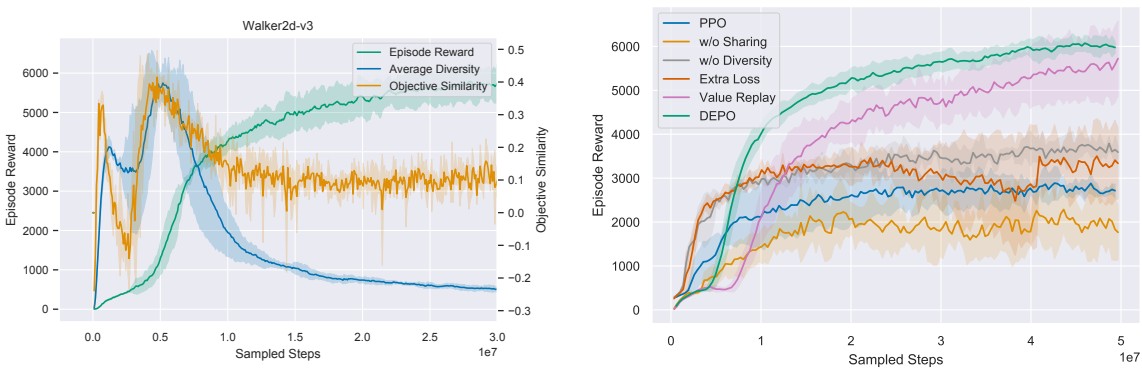

Figure 5: The learning dynamics of DEPO.

Figure 6: Ablation studies on the major mechanisms in DEPO.

## 4.3 Learning Dynamics

We investigate the dynamics of ensemble diversity in the course of training. As shown in Fig. 5, we plot two curves, the *average diversity* and the *objective similarity*, alongside with the reward curve of DEPO in Walker2d-v3 environment. The *objective similarity* is the cosine similarity between the task gradient and the diversity gradient $\cos\langle\mathbf{g}_t, \mathbf{g}_d\rangle$, showing the alignment between two objectives. The average diversity is the mean of the diversity reward of all policies $\mathbb{E}\sum_{i=1}^{K} r_d^{(i)}/K$.

An interesting observation is that the diversity as well as the objective similarity peaks when the performance is improved with the highest speed. When the sampled step is in range of $3M$ to $10M$, the objective similarity achieves high value, indicating the task gradient and diversity gradient are aligned to head to the similar

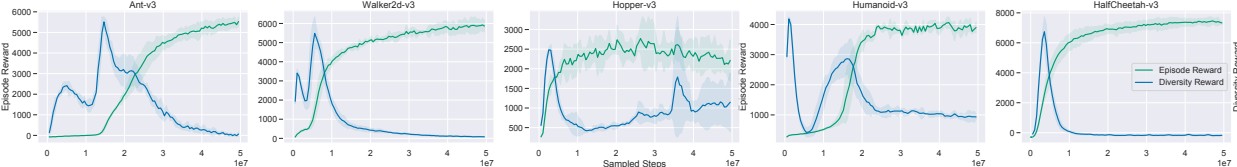

Figure 7: The tendency of diversity reward and episode reward. The diversity reward always peaks at the learning progressing drastically.

direction. This phenomenon suggests *in the early stage of training, finding diverse policies is finding better policies.* As shown in Fig. 7, the same phenomenon happens for all tested tasks. In the later training, the average diversity drops to low value and the objective similarity goes to zero. The policy ensemble becomes stable thus the diversity has a marginal impact to the policy improvement in the later stage of training. In short, we demonstrate empirically that the diversity regularization helps improving the exploration, especially in the early stage of learning.

### 4.4 Ablation Studies

To further understand which designed elements of DEPO are crucial, we conduct ablation studies thoroughly in Walker2d-v3 benchmark.

**Key components**. We first examine the key components of the proposed DEPO in Fig. 6. Compared to the baseline `PPO`, purely sharing data among the ensemble without diversity regularization (`w/o Diversity`) already boosts the performance. This result confirms the previous discovery that diversity introduced by different random initializations can promote performance (Osband et al., 2016).

On the contrary, disabling data sharing (`w/o Sharing`) decreases the performance significantly even if the diversity regularization is active. This is because data sharing broadcasts the experience of all policies so that all policies can optimize toward the high reward region collectively. When disabling data sharing, maximizing the diversity reward becomes the most feasible local minima for each policy, especially when the reward supervision from primal task is not significant in the early stage of training.

We also test the idea proposed in (Hong et al., 2018) (`Extra Loss`) to justify our usage of FDM as the gradient fusion method. We use an adaptive multiplier to balance the weighted sum of task and diversity objective: $\mathbf{g}_{\text{final}} = \mathbf{g}_t + \beta \mathbf{g}_d$, wherein $\beta$ increases by 0.05 if current policy's average diversity is lower than the running average for past 100 iterations and vice versa. However, due to the difficulty in tuning the trade-off between task and diversity, such method performs poorly compared to the DR.

Notice that in Eq. 10, we use the estimator $\hat{A}_{\pi_i}$ to estimate the advantage of policy $\pi_k$ in the state-action space sampled by the behavior policy $\pi_i$ instead of using the estimator $\hat{A}_{\pi_k}$ of policy $\pi_k$. Though we discuss that DEPO objective approximate the peer pressure objective through Theorem 1, we also verify this statement in the `Value Replay` experiment: we replay each value network $\hat{v}_{\pi_k}$ on the shared training batch and compute the estimated advantage $\hat{A}_{\pi_k}$ in Eq. 2 based on the replayed values for each policy. Intuitively, replaying the values will provide more accurate estimation to the policy gradient. However, the empirical result suggests such method performs inferior to the DEPO objective. DEPO converges faster than `Value Replay` and achieves better final performance.

**Impact of the ensemble size**. We reveal the impact of the ensemble size to the final performance. Note that when $K = 1$, the policy computes diversity against the delayed update target of itself. As shown in Table 2, the performance is improved when the ensemble size increase. However, when the ensemble size exceeds some threshold, preserving diversity will jeopardize the learning because one or more policies would learn diverse but weak behaviors, dragging down the whole training process. Finding an appropriate ensemble size can maximize the effectiveness of DEPO.

**Impact of the aggregation methods**. In Table 3, we examine different aggregation methods on policy ensemble. `Median` and `Elite` refer to the method using a single policy in the ensemble to sample action. The

| Table 2: Ensemble Size | |
|---|---|
| Size | Performance |
| K=1 | 2564.7 ±667.3 |
| K=3 | 5766.0 ±363.7 |
| K=5 | 6179.1 ±271.6 |
| K=10 | 6086.7 ±602.7 |

| Table 3: Ensemble Method | |
|---|---|
| Method | Performance |
| Median | 5660.3 ±590.1 |
| Elite | 6179.1 ±271.6 |
| Voting | 5710.1 ±344.1 |
| Average | 6379.4 ±241.1 |

| Table 4: Design Choice | |
|---|---|
| Ablation | Performance |
| w/o $\hat{A}$ Norm. | 1105.1 ±609.3 |
| w/o DVN | 2701.8 ±235.7 |
| w/ Ada. Div. | 2806.1 ±521.8 |
| w/ Mask | 4558.5 ±1129.6 |
| w/o TSC | 4624.4 ±796.2 |
| w/o DU | 4672.8 ±836.5 |

policy is selected based on its performance in evaluation and we select the median or the best policies. On the other hand, `Voting` refers to taking the action from policies that yields highest value: $a_t \sim \pi_i, i = \arg\max_i v_i$. The results shows that `Average` the output distributions yields the greatest performance gain and can even outperforms the best policy in the ensemble. More sophisticated aggregation methods, such as treating policy selection as a discrete RL problem, are left to future works.

**Other design choices**. In Table 4, we evaluate the significance of other design choices:

• `w/o` $\hat{A}$ `Norm.`: The advantage normalization demonstrates significant impact. Without such normalization, the diversity advantage and the task advantage might have disparate magnitudes, which imposes chaotic supervision to the learning and thus damages the performance.

• `w/o DVN`: The Diversity Value Network (DVN) also has huge impact. In this experiment, we disable the DVN and replace the diversity advantage with the discounted diversity return. However, the result suggests ablating DVN damages the training because the estimation of diversity return creates huge variance.

• `w/ Ada. Div.`: We use a simple heuristic to adapt the final gradient direction to justify the usage of angular bisector between two gradients. We adapt the $\beta$ when computing the direction of the final gradient $(1-\beta)\mathcal{Z}(\mathbf{g}_t)+\beta\mathcal{Z}(\mathbf{g}_d)$ in the same way as `Extra Loss` experiment. However, such method does not surpass the simple angular bisector. This setting does not limit the diversity gradients. On the contrary, the bisector bounds the diversity gradient projection so that its impact to the final gradient will not exceed the task gradient.

• `w/ Mask`: An technique to increase training diversity is also tested: during sampling, we generate a binary mask on each sample for each policy, and then filter the training batch for each policy according to the mask (Osband et al., 2016). By doing this, each policy will train on different data. However, similar to the finding in (Lee et al., 2020), this method reduces the training performance since the total data used to train each policy is reduced.

• `w/o TSC`: We find that using the Two-side Clip Loss proposed in Eq. 18 can further improve the performance.

• `w/o DU`: The Delayed Update target of policies can stabilize the computing of diversity reward and therefore improve the result.

## 5 Conclusion

In this work we explore the implementation of ensemble method into the policy optimization in RL and develop an ensemble policy optimization method called DEPO. The proposed method requests a set of policies to explore the environment simultaneously, trains each policy with shared training batch, and maintains the diversity of the ensemble with diversity regularization. The performance in test time is greatly improved by aggregating the learned policies into an ensemble policy. Experimental results show that the proposed ensemble policy optimization method can substantially improve sample efficiency in continuous locomotion tasks compared to the on-policy single-policy optimization counterparts. Detailed ablation studies reveal that the data sharing among the ensemble and the diversity regularization significantly improve the performance.

Though we conduct experiment on off-policy setting and show DEPO can achieve comparable performance with the baseline it built on, the advantage of DEPO is insignificant. The potential of combining policy ensemble with off-policy learning algorithm requires further exploration.

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

## A  Hyper-parameters

Table 5: Environment-related hyper-parameters of PPO and on-policy DEPO. The hyperparameters are selected based on PPO's performance.

| Parameter | H.C. | Ant | Walker | Hopper | Human. |
|---|---|---|---|---|---|
| LR | 0.0003 | 0.0001 | 0.0002 | 0.0001 | 0.0001 |
| $\lambda$ | 0.95 | 0.95 | 1.0 | 1.0 | 0.95 |
| SGD Epochs | 30 | 20 | 20 | 20 | 20 |

Table 6: Environment-agnostic hyper-parameters of SAC and off-policy DEPO. The hyperparameters are selected based on SAC's performance.

| Hyper-parameter | Value |
|---|---|
| Number of Agents | 5 |
| $\gamma$ | 0.99 |
| Training Batch Size | 256 |
| Maximum Steps | $1 \times 10^6$ |
| Steps that Learning Starts | 10000 |
| Learning Rate | 0.0003 |
| Delayed Update Coefficient ($\tau$) | 0.005 |

Table 7: Environment-agnostic hyper-parameters of on-policy DEPO.

| Hyper-parameter | Value |
|---|---|
| Number of Agents | 5 |
| KL Coefficient | 1.0 |
| Discount Factor $(\gamma)$ | 0.99 |
| Delayed Update Coefficient $(\tau)$ | 0.005 |
| Max Norm of Gradient | 10.0 |
| Use Diversity Value Network | True |
| Use Normalized Advantage | False |
| Use Delayed Update | True |
| Use Two-side Clip Loss | True |
| Maximum Sampled Steps | $5 \times 10^7$ |
| SGD Minibatch Size | 1024 |
| Training Batch Size | 10,000 |
| Number of Parallel Workers | 10 |
| Sampled Batch Size per Worker | 200 |

## B  Off-policy DEPO

To explore the applicability of DEPO in off-policy setting, we also implement an off-policy variant of DEPO using the maximum entropy framework based on Soft Actor-critic (SAC) (Haarnoja et al., 2018). Compared to SAC, we use an ensemble with $K$ policies to fill $K$ replay buffers $\{\mathcal{B}_i\}$. During training, we sample $N/K$ samples from each replay buffer and form a shared data batch with $N$ samples. Apart from $K$ Q networks $\{\hat{Q}_i\}_K$ to fit the Q values of $\pi_i$ on the shared data batch, additional $K$ diversity Q networks are used to compute the diversity gradient. The task objective for each policy can be written as:

$$J_{\text{Off}}(\pi_k) = \frac{1}{K} \sum_{i=1}^{K} \mathbb{E}_{(s,a) \sim \mathcal{B}_i} [\hat{Q}_k(s,a) - \alpha \log \pi_k(a|s)]. \tag{22}$$

We compare DEPO with three sets of off-policy baselines:

- `Off-policy`: We compare two popular off-policy and sample efficient methods: `SAC` (Haarnoja et al., 2018) and `TD3` (Fujimoto et al., 2018).

- `Exploration`: We benchmark an off-policy exploration-enhanced method: `OAC` (Ciosek et al., 2019). OAC further integrates SAC with the Upper Confidence Bound heuristics to conduct more informative exploration.

- `Ensemble Method`: We compare two ensemble methods with multiple policies: `ACE` (Zhang & Yao, 2019) and `SUNRISE` (Lee et al., 2020). Both works utilize a mixed policy to explore and train in the off-policy setting.

As shown in Fig. 8, off-policy DEPO converges faster than SAC which it built upon in three environments. Table 8 shows off-policy DEPO achieves comparable performance compared to SAC and TD3 in Hopper, Humanoid and Walker environment. Ant-v3 and HalfCheetah-v3 environments have loose constraints on action and huge action spaces, thus the diversity is more complex than other environments. Off-policy DEPO performs worse in these environments due to the difficulty in learning the diversity Q networks.

Off-policy DEPO is inferior possibly due to the extrapolation error of the Q functions. Compared to single agent method training Q function with whole data batch with $N$ samples, we sample only $N/K$ samples from each constituent policy $\pi_i$'s replay buffer. However, each Q function $\hat{Q}_i$ still need to estimate the Q values for $\pi_i$ in all $N$ samples. Therefore the estimation error of Q functions might increase compared to the single policy method.

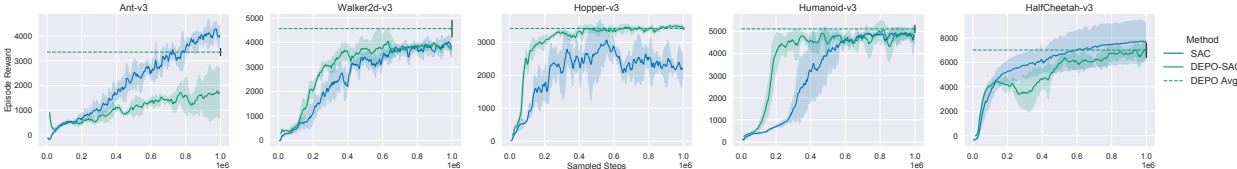

Figure 8: The learning curves of SAC and off-policy DEPO in five MuJoCo environments. DEPO achieves comparable performance with faster convergence in three tasks.

Table 8: Comparing off-policy DEPO with off-policy baselines.

| Category | Method | Ant-v3 | HalfCheetah-v3 | Hopper-v3 | Humanoid-v3 | Walker2d-v3 |
|---|---|---|---|---|---|---|
| Off-policy DEPO | DEPO Average | $3270.5_{\pm268.2}$ | $6994.5_{\pm679.1}$ | $3425.8_{\pm24.8}$ | $5183.3_{\pm123.3}$ | $4574.7_{\pm348.4}$ |
| | DEPO Elite | $1495.2_{\pm121.2}$ | $6340.8_{\pm305.6}$ | $3048.0_{\pm154.0}$ | $4776.8_{\pm193.6}$ | $3652.1_{\pm145.4}$ |
| Off-policy Baseline | SAC | $4654.2_{\pm272.3}$ | $7763.5_{\pm1777.4}$ | $3372.7_{\pm95.5}$ | $5021.3_{\pm165.9}$ | $4213.3_{\pm169.9}$ |
| | TD3 | $5143.5_{\pm436.6}$ | $10242.9_{\pm1149.4}$ | $3484.9_{\pm204.6}$ | $5220.1_{\pm81.0}$ | $4364.6_{\pm333.0}$ |
| Exploration | OAC | $4891.7_{\pm184.4}$ | $7576.6_{\pm499.0}$ | $3418.0_{\pm235.8}$ | $5128.8_{\pm78.0}$ | $3867.6_{\pm1126.8}$ |
| Ensemble Method | ACE | $1280.7_{\pm165.0}$ | $4131.2_{\pm330.8}$ | $1724.3_{\pm776.9}$ | $1529.3_{\pm493.2}$ | $3383.0_{\pm421.9}$ |
| | SUNRISE | $3902.0_{\pm1019.4}$ | $6518.3_{\pm1717.4}$ | $3639.1_{\pm90.2}$ | $5534.2_{\pm97.5}$ | $4981.1_{\pm982.0}$ |

