# OpenReview forum: "Ensemble Policy Optimization with Diversity-regularized Exploration"
_TMLR — Rejected by TMLR_

### Review · Reviewer_Kgdz · 2022-04-28

**Summary Of Contributions:**

The paper studies the problem of how to combine ensemble methods and RL, where authors investigate how to use an ensemble of policies to facilitate exploration in reinforcement learning. Authors propose the diversity-regularized ensemble policy optimization method, which trains multiple policies by having each of them to interact with environments to collect experiences (which are shared later). Authors use importance sampling to ensure stable data sharing for PPO. Authors also propose a diversity regularize to maintain the diversity of the policy ensemble. Experiments are conducted on MuJoCo tasks by applying the method to PPO.

**Broader Impact Concerns:**

No.

**Requested Changes:**

Please address the following parts as discussed above:
- Include more recent works in the discussion.
- Clarification for the opposite gradient direction case.
- More fair experimental comparison.

**Strengths And Weaknesses:**

## Strengths
- Relevance: The topic that the paper studies is relevant, which proposes to combine ensemble methods and RL for improving exploration and sample efficiency.

- Significance: Authors conduct experiments on several MuJoCo tasks, and achieve better results on most of the tasks.

- Novelty: Authors propose a peer pressure objective to motivate each agent to maximize its expected return and simultaneously outperform others in the ensemble. Authors also apply a two-side clip surrogate objective to prevent unbounded objectives following previous works. To maintain diversity, the authors propose to use diversity regularization by augmenting the original reward with a diversity reward. Overall, the proposed method seems to be interesting as far as I am concerned.

## Weakness
- How to combine ensemble methods with RL have rich literature, and I think the paper could be improved by discussing more related works such as EDAC, which also uses an ensemble of networks and also proposes diversity in the ensemble.

- In section 3.5, I am concerned about what would happen if the two gradient directions are opposite and how can the algorithm handle this case. Could authors better clarify this point?

- Regarding the soundness of experiments, I am a bit concerned about the fairness. DEPO uses multiple networks and a larger number of parameters, but most of the baselines have a significantly smaller number of parameters, which could weaken the fairness of experimental comaprison.

---

### Review · Reviewer_ExG4 · 2022-04-29

**Summary Of Contributions:**

The authors propose a framework for applying ensemble methods in RL. Key techniques in the framework are shared data, a novel peer pressure objective, a novel diversity regularization, and the use of feasible direction method for combining gradients from two different objectives. The authors experimentally study this framework with PPO and provide ablation study for the main techniques used in the framework.

**Requested Changes:**

Fixing the weakness 1- 5 is necessary for me to recommend acceptance.

**Strengths And Weaknesses:**

Strengths:
The paper is well written and easy to follow

Weaknesses:
1. The peer pressure objective is not properly motivated. I cannot not see why we need this new objective. What if we just optimize the original objective? There is no theoretical analysis showing the advantage of this new objective over the canonical one. There is no ablation study confirming the advantage empirically either.
2. The report of the empirical results are very misleading. (a) In Table 1, for Humanoid and Hopper, if we take error bars into consideration, the proposed algorithms and TD3 are equally good. However, only the score from the proposed algorithms are highlighted. This corresponds to the 5 random seeds issue raised by the action editor. Could the author clarify the protocol for highlighting? (b) If I understand it correctly, the on-policy algorithms all have 5e7 environment steps. But from Table 7, it seems off-policy algorithms have only 1e6 environment steps. Could the authors clarify this explicitly? It's fine that on-policy and off-policy algorithms have different steps. But then they shouldn't be put in the same table as Table 1.  If this is true, I do not think the claim "DEPO outperforms all on-policy baselines and achieves competitive performance compared to powerful off-policy, exploration and ensemble method baselines" is proper.
3. Off-policy setting is not properly investigated. As a framework for using ensemble in RL, I would expect it to work for both on-policy and off-policy algorithms. I'm happy to see that the authors indeed have an off-policy variant in Eq (19). But it looks this off-policy variant of DEPO is never empirically studied. I believe a comparison of this off-policy DEPO with other off-policy baselines are necessary for the publication of this work.
4. The comparison seems unfair. Table 6 reads that the proposed algorithms have task-dependent hyper parameters. Could the author clarify whether the baseline algorithms have task-dependent hyper parameters or not? If they have, those task dependent hyper parameters should also be listed.
5. Please remove random seed from Tables 7 & 8. Random seed is NOT a hyper parameter. This again goes back to the issue regarding the 5 runs raised by the action editor.

---

### Review · Reviewer_fib1 · 2022-05-03

**Summary Of Contributions:**

The paper introduces a new algorithm called Diversity-regularized Ensemble Policy Optimization (DEPO). The algorithm, trains multiple policies simultaneously (sharing data). Diversity is maintained via an additional objective that the authors refer to as "peer pressure". At test time, a mixtures of all individual policies is executed. The new algorithm is evaluated on 5 continuous control tasks. The paper also reports a number of ablation experiments and perform further analysis.

**Broader Impact Concerns:**

No concerns.

**Requested Changes:**

essential changes:
* I would avoid talking about exploration (certainly in the title) and clearly set out the problem the paper is addressing (i.e. policy optimization).
* Please avoid overclaiming and consider whether some of your results may be the result of random chance. More runs would be beneficial and I would like to see more careful statistical analysis. In particular you should not bold the highest performing method in tables if it is within one standard error of other methods. And some of the claims in the text are not supported by sufficient evidence.
* I am slightly concerned about the results in Figure 4. I would expect PPO to be able to solve Hopper-v3 which suggests sub optimal hyper parameters to me.
* You should cite the source of the tasks, not just Mujoco itself.
* page 2: $\max_a$ -> $\arg \max_a$
* $P_{\pi_\theta}$ should be introduced in (1)
* page 6, just before 3.4, I don't follow the argument for why D naturally decreases. Please strengthen this part.
* The objective is a reasonable heuristic but to my knowledge the wasserstein distance between two gaussians (even with diagonal covariance) has a term involving covariances (sec. 3.5).

recommended:
* the caption of Table 1 could mention the number of transitions.
* In Figure 4, I was surprised by the fact that DEPO basically flatlines for 1.5e7 steps on ant-v3. Could the authors comment on this effect?
* page 1, paragraph 2: I don't actually think the result referenced in the text is surprising. I would suggest removing this word.
* page 2: I imagine there's a better reference for UCB that's not from 2017.
* page 2: "on the contrary" -> "in contrast".
* While clearly written and easily comprehensible, the paper could benefit from some further proofreading. E.g. there are some articles missing throughout the paper.
* page 6: I don't think you need the off-policy objective in the paper at all, but if you want to keep it I suggest moving it to the ablation section.
* I'm not sure the WGAN reference is appropriate for the Wasserstein metric.

**Strengths And Weaknesses:**

Strengths:
* The paper has promising empirical results.
* To the extent that there are derivations I believe they are technically correct.
* The design choices are carefully ablated through a number of experiments. Furthermore the authors attempt to analyze the learning dynamics by looking at the diversity of constituent policies throughout training and an objective similarity metric that tracks the cosine distance between the gradient of the RL objective and the diversity objective.
* The paper is easy to follow and clearly written.

Weaknesses:
* The paper is framed in an odd way. The title suggest an exploration method but the algorithm and evaluation is on tasks that don't particularly require exploration in the conventional sense. Instead the algorithm, as far as I can tell uses an ensemble to collect diverse data and potentially avoid premature convergence to a sub-optimal solution.
* If the paper proposes a general method that claims to generally improve on PPO I think an evaluation in a wider set of tasks (e.g. not only continuous control) and settings would make for a stronger paper.
* I have some concerns (also raised by the AC) about the statistical significance of the results and overclaiming in the paper.
* There are minor parts of the paper that I believe are wrong (or should be clarified)

---

### Author Response · Authors · 2022-05-10
**Thanks to action editors and reviewers**

Dear action editors and reviewers,

Thank you for your effort in the review process of our submission! We already revised our submission subject to your suggestions and we briefly summarize the changes here:

1. We introduce the off-policy version of DEPO and empirically shows that it can accelerate the convergence of Soft Actor Critic (SAC), an off-policy baseline. The result is presented in **Section 4.4 Off-policy DEPO**.
2. We revise some typos, rephrase some sentences and add more discussion.

We are conducting large scale experiments repeating DEPO and important baselines with sufficient repeats. We will update our submission soon once those experiments are ready.

Thank you again for helping us improve the quality of our work!

Best,

DEPO authors

---

> ### Author Response · Authors · 2022-05-10
> **We also upload the code**
>
> We also upload the code of On-policy DEPO and Off-policy DEPO. Please find our code and relevant anonymized documentation in supplementary material. Thanks!

---

### Decision · Action_Editors · 2022-06-13

**Recommendation:** Reject

**Comment:**

The authors and reviewers had extended discussions. I appreciate the high level of engagement from the authors. The reviewers gave in-depth and appropriate evaluations. After the discussion, the paper was improved on several fronts, but some correctness issues linger. I encourage the authors to continue to improve these and consider resubmitting; I would be happy to re-review the work.

The primary issue here might be that the work is trying to do too much. One example might be the off-policy results. One reviewer sensibly mentioned removing the off-policy results, if they are not yet ready, and scaling back claims so that the algorithm is only for the on-policy setting. If this scaling back allows the authors to more carefully understand the algorithm, then this is the right decision. However, unclear off-policy results shouldn’t just be included in the appendix. Instead, they should be omitted. The whole paper should be correct and clear, including the appendix.

There are many choices in this work that should be carefully considered, as highlighted by the reviewers.

1. Framing as exploration.
The problem setting remains unspecified in the paper. Consider adding a section called Problem Formulation to clearly lay out the setting (one with a simulator where you can run policies in parallel) and that the goal is to improve exploration (data gathering).
Further, what is it about existing strategies that encourage exploration (like EDAC) that is still lacking, and how does your method fill that gap? A simple idea is good, but it should be solving a clear problem.

The work is currently focused on the method, namely using ensembles in RL. This motivation is ok, but on its own it is not enough. It is key to explain what problem is being solved, and the motivate why ensembles are well-suited to solve it.

2. The significance of the results.
It is good to get more runs for significance, but the point here was to ask: do I have sufficient evidence for the claims? Simply adding more runs is not the same as explaining clearly why the experiments provide evidence for the claims.

A part of this is making appropriate choices. There were some concerns about hyperparameter selection. For example, it is not clear why maximal performance is used for PPO to select hyperparameters, especially if you are evaluating the algorithm over the entire learning curve and evaluating online performance. How does this fit into your empirical question?

3. The off-policy experiments.
This was already mentioned above. But it is clear from the discussion with one reviewer that these need more work, or should be omitted.